# CO$_2$ Hydrogenation to Methanol by a Liquid-Phase Process with Alcoholic Solvents: A Techno-Economic Analysis

**Harri Nieminen \*, Arto Laari and Tuomas Koiranen**

Laboratory of Process Systems Engineering, Lappeenranta-Lahti University of Technology, P.O. Box 20 FI-53851 Lappeenranta, Finland

\* Correspondence: harri.nieminen@lut.fi; Tel.: +358-40-7451800

**Abstract:** Synthesis of methanol from recirculated CO$_2$ and H$_2$ produced by water electrolysis allows sustainable production of fuels and chemical storage of energy. Production of renewable methanol has, however, not achieved commercial breakthrough, and novel methods to improve economic feasibility are needed. One possibility is to alter the reaction route to methanol using catalytic alcoholic solvents, which makes the process possible at lower reaction temperatures. To estimate the techno-economic potential of this approach, the feasibilities of the conventional gas-phase process and an alternative liquid-phase process employing 2-butanol or 1-butanol solvents were compared by means of flowsheet modelling and economic analysis. As a result, it was found that despite improved methanol yield, the presence of solvent adds complexity to the process and increases separation costs due to the high volatility of the alcohols and formation of azeotropes. Hydrogen, produced from wind electricity, was the major cost in all processes. The higher cost of the present, non-optimized liquid-phase process is largely explained by the heat required in separation. If this heat could be provided by heat integration, the resulting production costs approach the costs of the gas-phase process. It is concluded that the novel reaction route provides promising possibilities, but new breakthroughs in process synthesis, integration, optimization, and catalysis are needed before the alcoholic solvent approach surpasses the traditional gas-phase process.

**Keywords:** CO$_2$ hydrogenation; methanol synthesis; liquid-phase process; alcohol promoted; process simulation; techno-economic analysis

## 1. Introduction

The synthesis of liquid fuels from hydrogen using captured CO$_2$ as the carbon source would allow sustainable fuel production with the potential to reduce CO$_2$ emissions in the energy and transportation sectors [1], while simultaneously providing an option for the chemical storage of intermittent renewable electricity [2]. Such an approach could potentially make a significant contribution to decarbonization of the energy system [3]. Methanol provides an example of such a liquid energy carrier [4].

Methanol is both an important industrial chemical and a useful multi-purpose fuel [5]. It can also be readily converted into products such as gasoline in the methanol-to-gasoline process (MTG) [6] or olefins in the methanol-to-olefins process (MTO) [7]. At present, most methanol comes from the catalytic conversion of synthesis gas (syngas) that is usually generated by steam reforming of natural gas [8]. The syngas, a mixture of hydrogen, CO, and CO$_2$, is converted into methanol on copper and zinc oxide (Cu/ZnO)-based catalysts at temperatures of 200–300 °C and pressures of 50–100 bar. The methanol synthesis process can be described by three equilibrium reactions:

$$CO_2 + 3H_2 \rightleftharpoons CH_3OH + H_2O \qquad \Delta H^0 = -49.8 \text{ kJ/mol} \qquad (1)$$

$$CO_2 + 2H_2 \rightleftharpoons CH_3OH \qquad \Delta H^0 = -91.0 \text{ kJ/mol} \qquad (2)$$

$$CO + H_2O \rightleftharpoons CO_2 + H_2 \qquad \Delta H^0 = -41.2 \text{ kJ/mol} \qquad (3)$$

Equations (1) and (2) represent the exothermic hydrogenation of $CO_2$ and CO to methanol, and Equation (3) represents the water-gas shift (WGS) reaction that is activated by the copper-based methanol synthesis catalysts [8]. As Reactions (1) and (2) are exothermic and result in a reduction of molar volume, methanol synthesis is favored at low temperatures and high pressures. However, sufficiently fast reaction kinetics requires temperatures above 200 °C, and methanol conversion is thus limited by the thermodynamic equilibrium.

Alternative to syngas, methanol can be produced by directly hydrogenating pure $CO_2$ with $H_2$ with high selectivity on conventional Cu/ZnO-based catalysts. However, the reaction rates are lower than with syngas feeds [9]. The equilibrium conversions are also lower compared to CO hydrogenation [10]. In addition to the thermodynamic limitation, methanol synthesis from pure $CO_2$ is complicated because of the increased water formation. In the absence of CO, water is produced both as the by-product of $CO_2$ hydrogenation (Equation (1)) and by the reverse-water gas shift reaction (reverse of Equation (3)). The increased formation of water leads to kinetic inhibition [11] and accelerated deactivation [12] of the Cu/ZnO catalysts.

The economic feasibility of $CO_2$ hydrogenation to methanol has been explored in a number of studies. While some studies paid close attention to the design and modelling of the methanol synthesis process [13–16], others focused on the electrolysis technology [17], electricity sources [18,19], or grid-scale implementation in a future renewable-based energy system [20]. Some studies have considered sustainability and environmental metrics in more detail [21,22]. Comparisons of methanol against other alternative energy carrier compounds have also been made [23]. Concerning the economics of the process, however, these studies draw significantly different conclusions. For example, Mignard et al. [13] and Anicic et al. [15] found the methanol production costs from $CO_2$ to be potentially competitive with fossil-based methanol production. In contrast, Pérez-Fortes [16] and Tremel et al. [23] found the production costs to be substantially higher than current methanol market prices. The overall costs have generally been found to be dominated by the hydrogen production costs, which consist of the electrolyzer capital costs and the cost of electricity.

There have been attempts to lower methanol processing costs by replacing the conventional gas-phase process with alternative liquid-phase processes. In the LPMeOH (liquid-phase methanol) process, the reaction is carried out in inert hydrocarbon solvent, allowing effective heat control of the exothermic reaction [24]. A demonstration-scale process has shown stable performance in conversion of coal-derived syngas with varying composition. Alternatively, methanol synthesis in co-catalytic alcoholic solvents has also been presented [25,26]. In the alcoholic solvent, methanol synthesis proceeds by an altered reaction mechanism via the formate ester of the alcohol, allowing lowered reaction temperatures. The lower temperature in turn allows higher equilibrium conversion in methanol synthesis.

The kinetics of the alcohol-promoted methanol synthesis process has been widely studied at the laboratory scale [25,26]. However, the techno-economic potential of this novel process has not been thoroughly examined. The aim of the present study is to assess the techno-economic feasibility of the liquid-phase alcohol-based process of $CO_2$ hydrogenation to methanol. For this purpose, the alcohol-promoted process with two alternative solvents is compared to the gas-phase process by means of process flowsheet simulation and subsequent economic analysis. 2-Butanol was selected as the primary solvent due to the good catalytic performance shown in experimental studies [27,28]. However, 1-butanol was also considered to assess whether a higher solvent boiling point would be favorable for the overall process efficiency and economics. It should be noted that published experimental details on the alcohol-promoted process are relatively limited, and the thermodynamics and kinetics have not been established in detail. Thus, the present work aims to provide a preliminary feasibility analysis rather than a rigorous optimization of the process alternatives. The key objectives

are to provide useful information for further development of the alcohol-promoted methanol synthesis process and to clarify its potential at the industrial scale.

## 2. Materials and Methods

Steady-state models of the processes for $CO_2$ hydrogenation to methanol were created in Aspen Plus (V9, AspenTech, Bedford, MA, USA). The processes studied included a gas-phase process and liquid-phase processes in alternative alcoholic solvents 2-butanol and 1-butanol. Mass and energy balances were generated and used to evaluate the technical performance of each process. The capital and operating costs of each process were estimated and compared and used to calculate the net present value (NPV) over the project lifetime. The boundaries of the present work are summarized in Figure 1. The design and costing of the $CO_2$ capture and water electrolysis units are outside the scope of the analysis, and the economic analysis was based on the referenced costs of $CO_2$ and hydrogen.

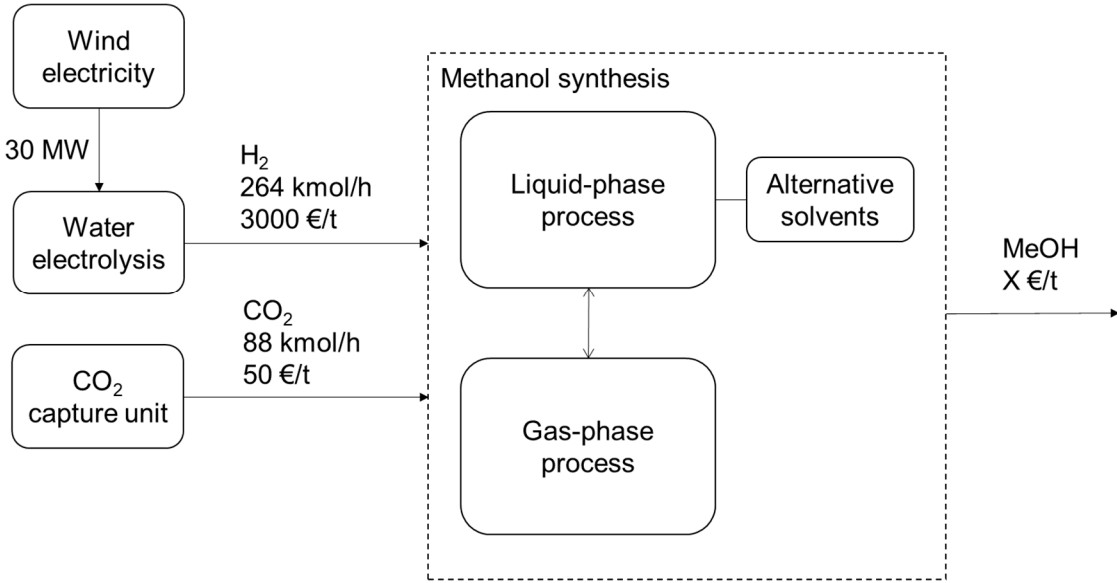

**Figure 1.** Scope and boundaries of the techno-economic analysis of $CO_2$ hydrogenation to methanol and the plant design capacity based on renewable electricity input.

### 2.1. Modelling Details

The capacity of the methanol synthesis unit is based on the amount of hydrogen available from the electrolysis unit powered by wind electricity at a 30-MW capacity. This capacity was selected as being representative of current wind energy projects in Finland [29].

The high-pressure sections (>10 bar) of each process were modelled in Aspen Plus using the RKSMHV2 (Redlich-Kwong-Soave with modified Huron-Vidal mixing rules) property method and the low-pressure sections using the NRTL-RK (Non-random two-liquid-Redlich-Kwong) property method. The property methods were selected following the guidelines given in Aspen Plus and taking into account the temperature, pressure, and polarity of the reaction system. All compressors were modelled with polytropic efficiency of 0.85 and mechanical efficiency of 0.95. Pumps were modelled at 0.85 pump efficiency and 0.95 driver efficiency. Heat exchangers were modelled by the shortcut method. The minimum temperature approach was set to 10 °C for liquid-liquid, 15 °C for gas-liquid, and 30 °C for gas-gas exchangers, and the pressure drop in each exchanger was set to 2% [30]. Distillation columns were modelled using the rigorous RADFRAC model in equilibrium mode.

The reactor in the gas-phase process was modelled using the RPLUG block model with an adiabatic setting. A relatively low inlet pressure of 50 bar was selected in order to facilitate comparison to the liquid-phase processes. The kinetics of the $CO_2$ hydrogenation to methanol and the water-gas shift reaction were estimated according to the model by Vanden Bussche and Froment [31] with readjusted

parameters by Mignard and Pritchard [32] and implemented in Aspen Plus as described by Van-Dal and Bouallou [14]. The reactor consisted of 1000 tubes with a length of 2 m and diameter of 0.05 m. The catalyst bed voidage was set at 0.4, particle density at 1775 kg/m$^3$, and particle diameter at 0.0055 m [14]. The pressure drop was calculated by the Ergun equation.

A bubble column reactor similar to the one utilized in the LPMeOH liquid-phase methanol synthesis process [24,33] was proposed for the liquid-phase process. The feed gases were bubbled through the solvent, and the product vapors together with unreacted gases were removed from the reactor. In the process with alcohol solvents, significant evaporation of the solvent took place, and the solvent vapors were removed together with the product vapors and gases. The solvent was then separated in downstream processing and returned to the reactor. Due to the lack of any detailed kinetic model for the alcohol promoted reaction route, the reactor in all liquid-phase processes was modelled with the RCSTR block based on the thermodynamic equilibrium by Gibbs energy minimization. In the model, $CO_2$ hydrogenation to methanol and the reverse-water gas shift were assumed as equilibrium reactions with the 0 °C approach to equilibrium. The reactor was operated isothermally at 180 °C and 50 bar, with reactions taking place in the liquid phase.

The sizing of the reactor for capital cost estimation was based on the specific methanol formation rate of 0.17 kg/(l h) reported by Tsubaki et al. for Cu/ZnO catalyst in ethanol solvent [34]. This rate was achieved in laboratory experiments under kinetics-controlled conditions with an approximate catalyst volume fraction of 1% in the slurry. In the present design, the same rate was assumed with a catalyst volume fraction of 10%, as limitations by mass and heat transfer are likely in a large-scale bubble column reactor. The same rate was also assumed regardless of the alcohol used as the solvent. The results of the reactor sizing are presented in Section 3.1.

## 2.2. Environmental Impact Analysis

The environmental impact of the alternative processes was assessed in terms of the $CO_2$ balance, electricity consumption, and water balance. In the calculation of the $CO_2$ balance, the amount of $CO_2$ fed to the process was subtracted from the sum of direct and indirect $CO_2$ emissions related to the process. These emissions consisted of $CO_2$ present in outlet streams, the $CO_2$ emitted in steam generation, and the indirect emissions of grid electricity. The specific electricity consumption (per t MeOH) of the processes was calculated, and the corresponding $CO_2$ emission was estimated from the carbon intensity of the Finnish electricity grid at the time of writing (170 g $CO_2$/kWh) [35]. For steam generation, emissions from both the combustion of externally-supplied fuel (natural gas) and the combustion of process waste streams were considered. Cooling water input and waste water output were considered in the water balance. The mass flow rate and composition of the waste water streams, consisting of water/alcohol mixtures, were assessed.

## 2.3. Cost Estimation

The following section describes the methods used and the assumptions made in the evaluation of the capital and operating costs of the $CO_2$ hydrogenation to methanol processes.

### 2.3.1. Capital Costs

The capital costs were estimated by the factorial method according to Towler and Sinnott [36]. The installed equipment costs for the estimation were obtained from the cost functions integrated into the Aspen Plus software. The installed costs in USD were converted to Euros at the exchange rate of 0.89 €/USD (2018). The installed costs were further corrected for construction from SS304 stainless steel by a material factor of 1.3 [36] and by a location factor of 1.043 corresponding to Western Europe [16]. The reactor's cost in the liquid-phase process was based on the sizing procedure described above. A 50% contingency was added on top of the cost of the pressure vessel in order to account for auxiliary equipment such as heat transfer equipment, slurry handling, and catalyst activation. The reactor cost

was identical in all the liquid-phase processes, as the effect of different solvents on the reaction rate and the resulting reactor volume was not considered.

The corrected installed equipment costs corresponded to the inside battery limits (ISBL) capital costs comprising the purchase and installation of all the main and auxiliary process equipment. The offsite (OSBL) capital costs, including the infrastructure and site improvements, were calculated as 25% of the ISBL capital costs. The plant would be preferably located on an existing fuel production site with readily-available infrastructure. Engineering costs were estimated as 20% of the sum of the ISBL and OSBL costs. Finally, a contingency of 30% of the sum of ISBL and OSBL costs was added to obtain the total fixed capital cost (TFCC). The working capital was estimated as 15% of the sum of the ISBL and OSBL costs. The factorial method of capital cost estimation is summarized in Table 1.

**Table 1.** Factorial method of capital cost estimation [16,36]. ISBL, inside battery limits; OSBL, offsite battery limits.

| Item | Basis |
|---|---|
| ISBL capital cost | Installed equipment cost from the Aspen Plus<br>Exchange rate of 0.89 €/USD<br>Material factor 1.3 (304 stainless steel)<br>Location factor 1.043 (Western Europe) |
| OSBL capital cost | 25% of ISBL |
| Engineering cost | 20% of ISBL and OSBL |
| Contingency | 30% of ISBL and OSBL |

To calculate its contribution to the total methanol production cost, the total fixed capital cost was annualized based on an assumed plant lifetime of 20 years and an interest rate of 5%.

### 2.3.2. Variable and Fixed Operating Costs

The overall cost of hydrogen production, including capital and operating costs of both the 30-MW wind farm and the alkaline electrolysis unit and the hydrogen storage costs, was assumed to be 3000 €/t of hydrogen. This value was based on a 2006 report by Levene et al. [37], which estimated that the production cost of wind-based hydrogen was in the range of $2.90–3.40/kg, including hydrogen storage. The production cost of wind electricity in the Finnish scenario has been recently estimated at 41.4 €/MWh [38]. This value is fairly consistent with the wind electricity cost ($0.038/kWh) used by Levene et al. [37]. The hydrogen cost is also consistent with Smolinka et al. [39], who estimated a value of 3.17 €/kg for large-scale alkaline electrolysis with intermittent operation (average capacity factor 35%). All the electricity available from the wind farm was utilized in the electrolysis unit. In order to maintain constant operation, the methanol synthesis unit was powered by grid electricity, available at an assumed market cost of 60 €/MWh [40]. Electricity consumption of the synthesis unit was calculated in the Aspen Plus process models.

The cost of $CO_2$ consisted of the capital and operational costs of an amine absorption unit. A cost of 50 €/t was assumed based on the International Energy Agency report [41]. If the $CO_2$ capture unit is located at a distance from the electrolysis and synthesis units, $CO_2$ transportation cost should also be included. However, this was not considered as the transport cost was small compared to the $CO_2$ capture costs [42].

The cost of steam was calculated based on a fuel (natural gas) cost of 30 €/MWh [43] and boiler efficiency of 80%, including heat losses. The fuel cost was calculated for the generation of medium-pressure (MP) steam at 20 bar (saturation temperature 212 °C). The overall cost was corrected by a factor of 1.3 taking non-fuel costs into account [44]. As a result, a cost of 35 €/t was obtained for the MP steam, and an identical cost was assumed for the low-pressure (LP) steam at 6 bar (saturation temperature 159 °C). Shaft work or condensate credits were not considered. In process modelling, MP and LP steam were included as utilities in the Aspen Plus model for calculation of the steam

consumption rate. Full condensation of steam in exchangers was assumed, and the outlet temperatures of MP and LP steam were set at 211 °C and 158 °C, respectively.

Steam generation by waste heat available from the combustion of process waste and purge streams was also considered. Both gas/vapor and liquid streams suitable for combustion were included in waste heat generation. Lower heating values of 10.1 MJ/kg for CO, 121 MJ/kg for $H_2$, 19.9 MJ/kg for methanol, and 34.4 MJ/kg for both 2-butanol and 1-butanol were used in the calculation of the heat produced [45]. A boiler efficiency of 80% was assumed for the waste heat boilers. The steam generated by the waste heat was utilized in the processes by subtraction of the amount of steam generated from the process MP steam consumption. In cases where the process produced a net heat output, the steam generated was considered a by-product with a selling price of 35 €/t.

The consumption rate of cooling water was also calculated in the Aspen Plus process models. The cost of cooling water was 0.26 €/m$^3$ [42], with an inlet temperature of 20 °C and outlet temperature of 25 °C. The cost of waste water was 0.32 €/m$^3$ [42] regardless of the composition of the waste water streams. Consumables included the methanol synthesis catalyst (assumed cost of 95 €/kg [16] and lifetime of 4 years) and the solvent make-up. The amount of catalyst used in the gas-phase process (3.49 t) was calculated based on the volume of the reactor tubes (1000 tubes, length 2 m, diameter 0.05 m), catalyst density (1775 kg/m$^3$ [14]), and bed porosity (0.5). Assuming a 4-year catalyst lifetime, 0.87 t of the catalyst needs to be replaced each year, giving a per year cost of approximately 83,000 €, which was not discounted. The amount and cost of catalyst used in the liquid-phase methanol synthesis processes was calculated by the reactor sizing procedure described in Section 2.1. The cost of make-up solvent was assumed to be 500 €/t in the liquid-phase processes, regardless of the alcohol used. A summary of the variable costs considered is given in Table 2.

**Table 2.** Variable costs considered in the analysis. MP, medium-pressure; LP, low-pressure.

| Item | Cost and Details |
|---|---|
| Hydrogen | 3000 €/t, based on alkaline electrolysis powered by 30 MW of wind electricity (cost includes electricity production and hydrogen storage) [37,39] |
| Grid electricity | 60 €/MWh [40] |
| $CO_2$ | 50 €/t [41] |
| Steam | 35 €/t for MP (20 bar) and LP (6 bar) steam, based on natural gas cost of 30 €/MWh [43] |
| Cooling water | 0.26 €/m$^3$ [42] |
| Waste water | 0.32 €/m$^3$ [42] |
| Catalyst | 95.24 €/kg [16], assumed lifetime 4 years |
| Solvent make-up | 500 €/t for all alcohols |

Fixed operating costs were calculated according to the factorial method from Towler and Sinnott [46]. A labor requirement of 4 shift positions with 4 operators per position with a salary of 40,000 €/a was assumed. Supervision was estimated as 25% of labor cost. Labor overheads were assumed as 45% of the sum of labor and supervision. Maintenance costs were assumed as 3% of the ISBL capital cost. Plant and company overheads constituted 65% of the labor and maintenance costs, while taxes and insurance constituted 2% of the total fixed capital cost.

### 2.3.3. Revenues

A methanol price of 400 €/t [47] was assumed in the economic analysis. Additional revenues from the sales of oxygen by-product generated in the electrolysis unit were also considered. A conservative price of 70 €/t [15,18] was assumed for oxygen, and the costs of oxygen compression and liquefaction were omitted.

*2.4. Economic Analysis*

The net present value (NPV) of each process was calculated based on the following assumptions. Plant lifetime was set at 20 years. Thirty percent, 60%, and 10% of the capital costs were distributed to Years 1, 2, and 3, respectively. Thirty percent and 70% of revenues and operating costs were considered during Year 3 and Year 4, and 100% thereafter. One hundred percent of working capital was deployed during the first year. A discount rate of 8% was assumed, and taxes and depreciation were not considered [16].

*2.5. Process Descriptions*

The overall process discussed here consisted of the electrolysis unit powered by wind electricity and the methanol synthesis unit. Options for the methanol synthesis unit included the gas-phase synthesis process and the liquid-phase synthesis process with alternative solvents (2-butanol and 1-butanol).

2.5.1. Electrolysis and Wind Electricity

The present analysis considered wind electricity in the Finnish scenario [38] for the electrolysis process. The plant consisting of the electrolysis unit, the methanol synthesis unit, and possibly the $CO_2$ capture unit was assumed to be located near a land-based wind turbine farm. Transportation of $CO_2$ was not ruled out, as this location assumption might prove unrealistic. Potential sources of $CO_2$ would consist of fossil power plants and various industrial sources (especially bioprocessing plants in the Finnish scenario).

Based on available data on current and upcoming wind energy projects in Finland [29], the electricity generation capacity of the wind farm was set at 30 MW. Due to the significant temporal variation inherent in wind-based electricity generation, the full capacity was not constantly available for the electrolysis unit. However, a constant supply of hydrogen to the methanol synthesis unit is required to allow steady-state operation at design capacity. Thus, a sufficient capacity for hydrogen storage for the methanol synthesis unit should be assumed.

Hydrogen was generated by pressurized alkaline electrolysis operating at 30 bar. The capacity of the electrolysis unit was 30 MW, and the system efficiency was 70% [39]. Based on the heat of formation of water (285.8 MJ/kmol), 264.5 kmol/h of water was split to form an equal amount of hydrogen in moles. At the 30-bar operating pressure, this corresponds to 533.2 kg/h of hydrogen fed to the methanol synthesis unit.

2.5.2. Gas-Phase Methanol Synthesis

The flowsheet of the gas-phase methanol synthesis process is presented in Figure 2. At the feed compression stage, hydrogen was compressed from 30.0 bar (outlet pressure of the alkaline electrolyzer) to 51 bar in a single stage (COMP5), and $CO_2$ was compressed from 1.0 bar–51.0 bar in four stages (COMP1–4) with intercooling (COOLER1–3). The pressure ratio of Stages 1–3 equaled 3.0, while the final stage was specified to the outlet pressure of 51.0 bar. The molar ratio of the fresh feed was three moles hydrogen per one mole $CO_2$, with mass flows of 533.2 and 3881.7 kg/h, respectively. The feed gases were mixed with the recycle gas (MIX1), and the mixed feed was preheated to 215.0 °C by heat exchange with the reactor outlet in the heat exchanger HX1. The feed was then converted in the adiabatic reactor.

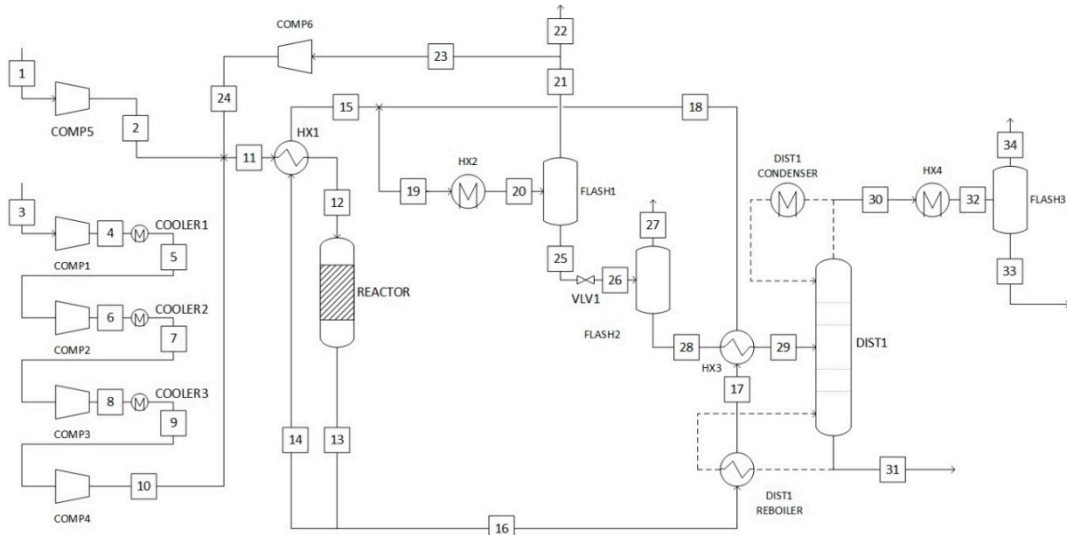

**Figure 2.** Flowsheet of the gas-phase $CO_2$ hydrogenation to methanol process. The corresponding stream table is presented in Table S1 (Supplementary Material). Inlet and outlet streams: 1. Hydrogen inlet, 3. $CO_2$ inlet, 22. purge from gas recycle, 27. gas purge from second flash tank, 31. waste water outlet, 33. methanol product outlet, 34. gas purge from final flash tank. COMP, compression stage; HX, heat exchanger; DIST, distillation column.

The temperature at the reactor outlet was 274.4 °C. The outlet gas was split into two fractions in SPLIT1, with 68% of the gas (Stream 14) used to preheat the feed in HX1. The remaining 32% was heat integrated with the reboiler of the distillation column and preheated the column feed (HX3). The two streams were recombined (MIX2) and cooled to 35.0 °C in the cooler HX2. The unreacted gases were separated in the flash drum (FLASH1). For the recycled gas, 1% was purged in order to avoid accumulation of by-products and inert components [14]. Such components were however not included in the process model. The remaining recycled gas was recompressed to 51.0 bar in COMP6 and mixed with the fresh feed.

The liquid separated in FLASH1 consisted of methanol and water at a molar ratio of approximately one, together with a small fraction of dissolved gases. The pressure of this stream was reduced to 1.2 bar, and the majority of the dissolved gases were separated and purged in FLASH2. The liquid stream was heated to 81.0 °C in HX3 and fed to the distillation column (DIST1) operated at 1.2 bar. The column consisted of 30 ideal equilibrium stages, and the reflux ratio equaled 1.1. The top product, consisting mainly of methanol and dissolved $CO_2$, was cooled to 35.0 °C in HX4, and most of the $CO_2$ was separated in FLASH3. The final purity of the methanol product was 99.3 wt%. Water (99.0 wt%) was removed from the bottom of the distillation column.

2.5.3. Liquid-Phase Methanol Synthesis

The liquid-phase methanol synthesis process was based on the combination of a conventional Cu/ZnO catalyst and alcohol as a catalytic solvent. In the presence of the alcoholic solvent, the reaction proceeded through the formate ester of the corresponding alcohol as an intermediate [25]. This reaction mechanism allowed methanol synthesis at lower reaction temperatures compared to the gas-phase process.

The flowsheet of the liquid-phase methanol synthesis process using 1-butanol solvent is presented in Figure 3. The liquid-phase process was also modelled with 2-butanol using identical process design. In the following description, the stream conditions and compositions of the 1-butanol process are used as examples. The major differences between the two solvents were found in the design and performance of the separation stage, as summarized in Table 3 at the end of this section.

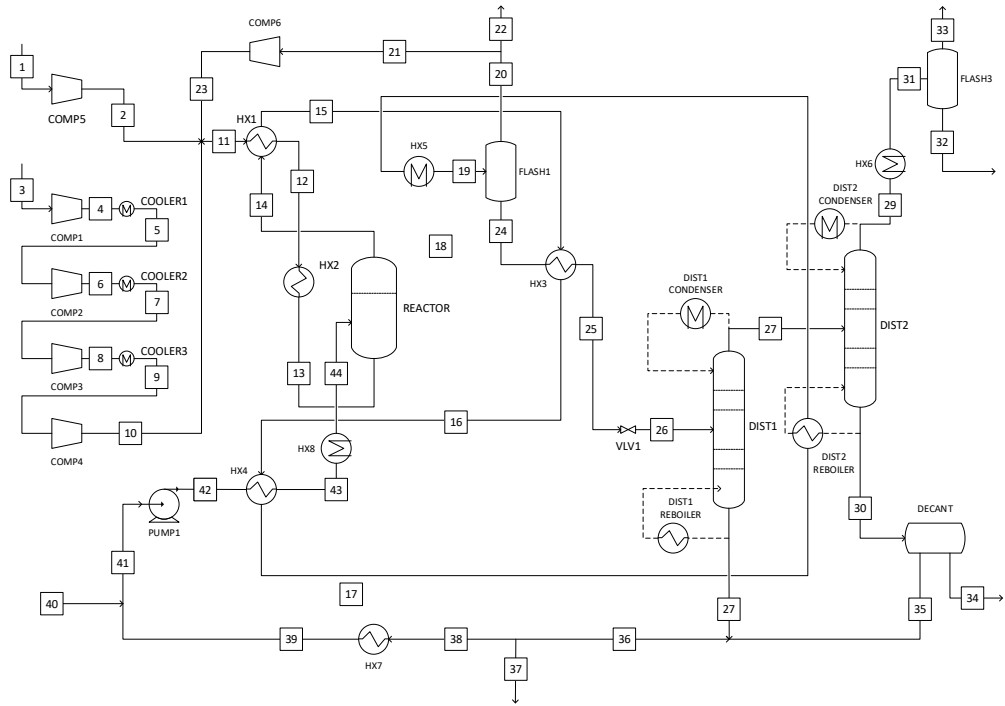

**Figure 3.** Flowsheet of the liquid-phase $CO_2$ hydrogenation to methanol process with 1-butanol solvent. The corresponding stream table is presented in Table S2. 1. Hydrogen inlet, 3. $CO_2$ inlet, 22. purge from gas recycle, 32. methanol product outlet, 33. gas purge from final flash tank. 34. waste water outlet, 37. purge from solvent recycle, 40. solvent make-up.

**Table 3.** Distillation specifications and performance in the alternative methanol synthesis processes.

|  | **Gas-Phase** | **2-Butanol** | **1-Butanol** |
|---|---|---|---|
| Distillation feed flow rate, kg/h | 3660 | 26,641.1 | 14,650 |
| **Column #1** | | | |
| Number of ideal stages | 30 | 15 | 10 |
| Reflux ratio (molar) | 1.1 | 1.0 | 1.0 |
| Reboiler duty, kW | 940 | 7282 | 5445 |
| **Column #2** | | | |
| Number of stages | - | 50 | 30 |
| Reflux ratio | - | 6.5 | 4.0 |
| Reboiler duty, kW | - | 4210 | 2265 |
| Methanol purity * (wt%) | 99.3% | 99.2% | 99.2% |

* Following FLASH3 (35 °C, 1 bar).

Identical to the gas-phase process, 533.2 kg/h of hydrogen and 3881.7 kg/h $CO_2$ were fed to the liquid-phase process. The hydrogen feed was compressed to 52.0 bar in a single stage (COMP5), and the $CO_2$ feed was compressed to 52.0 bar in four stages (COMP1–4) with intercooling (COOLER1–3). The pressure ratio of Stages 1–3 equaled 3.0, while the final stage was specified to the outlet pressure of 52.0 bar. The feed gases were mixed with the recycled gas (MIX1), and the mixed feed was preheated to 138.1 °C by heat exchange with the reactor vapor outlet (HX1). The feed was further heated to 180.0 °C in the heater HX2. The feed gas was fed to the isothermal bubble column reactor operated at

180.0 °C and 50.0 bar. Details on the design and modelling of the liquid-phase reactor are discussed in Section 2.1.

The unreacted gases together with product and solvent vapors were removed from the reactor, while the liquid level was maintained by the solvent recycled. The heat from the gas/vapor outlet (Stream 14) was used to preheat the reactor feed in HX1. The heat available in the outlet stream was utilized to pre-heat the distillation feed (HX3), to pre-heat the solvent recycle (HX4), and in the reboiler of the second distillation column (DIST2). The product stream was then further cooled to 35 °C in HX5, and the majority of the unreacted gases were then separated from the condensed solvent and products in FLASH1. In SPLIT1, 1% of the recycled gases was purged and the remainder recompressed to 52.0 bar (COMP6) and mixed with the fresh feed gases.

The condensed liquid stream leaving FLASH1 was pre-heated in HX3; the pressure was reduced to 1.4 bar, and the stream entered the first distillation column (DIST1) at 75.4 °C. Methanol and water were removed as the top product. A fraction of the solvent was also distilled due to the alcohol-water azeotrope. The majority of the solvent was removed from the bottom stage and recycled.

The vapor product from DIST1 was fed to the second distillation column (DIST2), operated at 1.2 bar. Methanol was removed from the top of the column, while water and the remaining solvent were removed from the bottom. The methanol stream was cooled to 35.0 °C in the exchanger HX6, and most of the dissolved $CO_2$ remaining was removed in FLASH3. The purity of the methanol product was 99.2% by weight. The mixture of water and solvent from the bottom stage was fed to a decanter at 95.8 °C and 1.2 bar. In the decanter, the heterogeneous alcohol-water azeotrope was split into a removed water-rich (89% by mass) waste stream and a solvent-rich (81% 1-butanol) recycled stream (2-butanol process: 90% water and 82% 2-butanol, respectively). The recycled gas was mixed with the solvent removed in DIST1, resulting in a final composition of 96 wt% 1-butanol and 4 wt% water in the solvent recycled. One percent of the solvent recycled was purged, and the stream was cooled in HX7 for vapor condensation. The recycled gas was then mixed with the solvent make-up, and the pressure of the mixed stream was then increased to the reactor pressure by the solvent pump (PUMP1). The solvent was pre-heated to 136.7 °C in HX4 and finally heated to 180 °C in HX8.

Details of the specifications and performance of the distillation columns in the gas-phase and liquid-phase processes are given in Table 3. The data allowed us to compare the simplicity of separating methanol from each product-solvent mixture. In the gas-phase process, only a single column with low energy input was required to separate methanol from the water by-product. In the liquid-phase processes, two columns were necessary to separate both methanol and water from the solvent. In 2-butanol, this separation was significantly capital and energy intensive due to the similar volatility of the components (2-butanol has a boiling point of approximately 99 °C). The separation was less costly with 1-butanol, which has a boiling point of 117.7 °C. The formation of azeotropes between water and the solvent was also a complicating factor in the separation processes. The azeotropic mixtures with water consisted of 40% of 2-butanol (at 87.2 °C) and 25% of 1-butanol (92.5 °C) on a molar basis [48].

The energy consumed in distillation could be reduced by more rigorous heat integration in the processes. In all of the processes, the heat from the hot reactor outlet was exchanged to preheat the distillation feed, based on the implementation of the gas-phase process by Van-Dal and Bouallou [14]. In the liquid-phase processes, alternative heat integration schemes could lead to energy savings, but this was not explored in detail in the present work.

## 3. Results and Discussion

The various methanol synthesis processes were compared in terms of the mass balances, energy and electricity consumption, and the overall methanol production cost. The performance of the reactor in the gas-phase and liquid-phase processes was also compared. Additional results with more details are included in the Supplementary Material, as referenced in the text.

### 3.1. Reactor Sizing and Performance

Figure 4 presents the temperature and composition profile of the adiabatic reactor in the gas-phase process. The composition is given in terms of the mole fractions of $CO_2$, CO, and methanol. The mole fraction of water over the reactor length was essentially identical to that of methanol. The reactions modelled here, i.e., $CO_2$ hydrogenation to methanol (Equation (1)) and the reverse-water gas shift (RWGS, Equation (3)), reached equilibrium after one meter of reactor length. Due to the low contribution of the reactor capital cost to the overall production cost (Section 3.4), this was considered satisfactory, and further reactor optimization was omitted. At this point, the peak temperature inside the reactor (274.5 °C) was reached, and 20.3% of the $CO_2$ entering the reactor was converted. Thus, a significant amount of unreacted gases was recycled, and the recycle ratio equaled 5.3 in the reactor loop. $CO_2$ was converted to methanol at a selectivity of 96.1%. The mole fraction of CO remained low throughout the reactor length, and the constant concentration of CO after approximately 0.5 m corresponded to the equilibrium level of the RWGS.

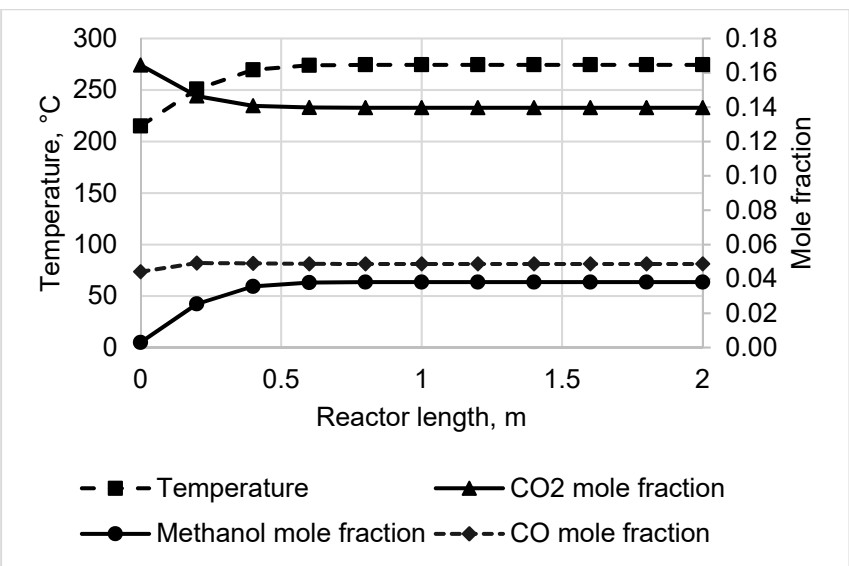

**Figure 4.** Reactor temperature and composition profile in the gas-phase $CO_2$ hydrogenation to methanol process. Adiabatic reactor, inlet temperature 215.0 °C, and pressure 52.0 bar.

The lack of a detailed kinetic model for the alcohol-promoted process prevented detailed reactor modelling in the liquid-phase processes. Therefore, the reactor was approximated as an ideal continually-stirred tank reactor (CSTR) with full equilibrium conversion of the $CO_2$ hydrogenation to methanol and RWGS reactions. This method gave a "best-case scenario" view of this process, as the reaction kinetics was ignored.

In the 1-butanol process, approximately 2.7 t/h of methanol was produced in the reactor based on the reactor mass balance. Based on the specific methanol formation rate of 0.17 kg/(l h), the slurry (liquid + catalyst) volume required was thus 15.8 $m^3$. At a 5% catalyst volume fraction, the volume of the catalyst was 0.8 $m^3$. At a catalyst density of 1775 kg/$m^3$ [14], the mass of the catalyst was 1.4 t. Assuming a 25 vol% provision for gas/vapor space, the total reactor volume was 17.8 $m^3$. To determine the aspect ratio, a superficial gas velocity of 0.1 m/s [49] was assumed, resulting in a reactor diameter of 1.7 m and a height of 8.8 m. The average residence time of the gas in the reactor was 67 s and the hourly space velocity per catalyst weight was 576.3 L/kg h and per slurry volume 51.1 L/h. The same reactor sizing was assumed for the 2-butanol process.

Table 4 presents the composition of the inlet and outlet streams to the reactor in the 1-butanol process. A similar distribution of reactants and products in the liquid and gas/vapor phases was found in other solvents. The gas-phase inlet consisted of the feed gas mixture, and the recycled and make-up

solvent constituted the liquid phase inlet. The gas/vapor outlet consisted of the product and solvent vapors together with unreacted gases.

**Table 4.** Composition of the gas/vapor and liquid phases in the reactor inlet and outlet streams of the liquid-phase process with 1-butanol solvent. Reaction conditions are 180.0 °C and 50.0 bar, and the reactor is modelled as a continually-stirred tank reactor (CSTR) with the full equilibrium approach of the $CO_2$ hydrogenation to methanol and the reverse-water gas shift reactions.

| Reactor Inlet | | | | Reactor Outlet | |
|---|---|---|---|---|---|
| Gas/Vapor Phase | | Liquid Phase | | Gas/Vapor Phase | |
| Component | Flow, kmol/h | Component | Flow, kmol/h | Component | Flow, kmol/h |
| $CO_2$ | 118.6 | $CO_2$ | 0.0 | $CO_2$ | 36.2 |
| CO | 0.8 | CO | 0.0 | CO | 0.8 |
| Methanol | 1.4 | Methanol | 0.4 | Methanol | 84.2 |
| Hydrogen | 934.9 | Hydrogen | 1.4 | Hydrogen | 689.0 |
| Water | 0.7 | Water | 23.1 | Water | 106.2 |
| 1-butanol | 0.2 | 1-butanol | 132.7 | 1-butanol | 132.8 |
| Total | 1056.5 | Total | 157.6 | Total | 1049.3 |

The lower reaction temperature (180.0 °C) compared to the gas-phase process led to a relatively higher $CO_2$ conversion. The per-pass conversion was 81% with 1-butanol and 79% with 2-butanol as the solvent. The phase distribution of the reactants and the products appeared to have a favorable effect on the conversion, as the single-phase equilibrium conversion was in the region of 40% at the present reaction conditions [10]. The evaporation of methanol and water from the liquid phase seemed to drive the conversion of the reactants. The slightly different phase distribution due to the higher volatility of 2-butanol was also suggested to lead to the different conversion value in the 2-butanol and 1-butanol processes.

*3.2. Mass and Energy Balances*

Table 5 gives a summary of the mass balances of the alternative methanol synthesis processes. More detailed balances with the individual inlet and outlet streams can be found in Tables S4–S6 in the Supplementary Material. The overall methanol yield was calculated as the mass flow of methanol in the product stream divided by the stoichiometric methanol output based on the $CO_2$ and hydrogen inlet. The yield was decreased by losses of reactants or methanol from the process. In Table 5, the methanol yield is 81% in the gas-phase process, 82% in the 2-butanol process, and 88% in the 1-butanol process.

**Table 5.** Mass balance of the alternative methanol synthesis processes. Methanol yield is the ratio of the mass flow of methanol product to the mass flow corresponding to 100% yield based on the stoichiometric $CO_2$ and hydrogen inlets. Fractional solvent loss is calculated as the ratio of solvent flow removed from the process to the solvent flow (make-up + recycle) fed to the reactor.

| Flow, kg/h | Gas-Phase | 2-Butanol | 1-Butanol |
|---|---|---|---|
| Hydrogen in | 533 | 533 | 533 |
| $CO_2$ in | 3882 | 3882 | 3882 |
| Methanol out | 2275 | 2311 | 2474 |
| Methanol losses | 44 | 248 | 167 |
| $CO_2$ losses | 560 | 369 | 255 |
| Hydrogen losses | 89 | 51 | 35 |
| $CO_2$ conversion per pass | 20% | 79% | 81% |
| Methanol yield | 81% | 82% | 88% |
| Solvent loss, kg/h | - | 352 | 249 |
| Fractional solvent loss | - | 2% | 3% |

Losses of $CO_2$, hydrogen, and methanol are also seen in Table 5. Components were lost with the purge and waste streams removed from the processes. In the gas-phase process, these streams consisted of the purge from the gas recycle, the bottoms from the distillation column, and the purged gases from the separators FLASH2 and FLASH3 (see the flowsheet in Figure 2). The majority of the methanol loss occurred in the purge from FLASH2. Due to the large amount of unreacted $CO_2$ and the hydrogen present in the recycle stream, the loss of reactants with the purge was significantly higher compared to the liquid-phase processes, leading to the lower overall methanol yield.

In the liquid-phase processes, components were lost with the gas and solvent recycle purges, the water-rich stream from the decanter, and also the purge stream from FLASH3 (Figure 3). The purge from FLASH3 constituted most of the methanol and gas losses, while the solvent was mainly lost from the decanter. This loss is explained by the alcohol-water azeotrope. A significant amount was also lost with the liquid purge stream, which could only be minimized based on detailed information on the formation and accumulation of by-products. Overall, the fraction of solvent lost was 2% for 2-butanol and 3% for 1-butanol. However, the absolute amount of solvent lost was higher in the 2-butanol process, as a larger amount of solvent was circulated in order to maintain the liquid/gas volume ratio in the reactor.

The presence of a large quantity of solvent in the product streams significantly increased the distillation energy requirement in the liquid phase process, especially as a second column was required. For each process, the overall heat and electricity consumption per ton of methanol produced are presented in Table 6. It should be noted that the electricity required in the electrolysis unit was included in the cost of hydrogen, and hence was not included here. Electricity consumption in Table 6 only includes that used in compression of the gaseous feed and recycle streams and in pumping the recycled solvent in the liquid-phase processes. The waste heat generated by combustion of the process purge and waste streams was also included in the energy balance. The hot utility requirement corresponded to the amount of external heat required after integration of the waste heat.

**Table 6.** Thermal electrical energy consumption (in kWh per t of methanol produced) of the alternative $CO_2$ hydrogenation to methanol processes.

| Energy, kWh/t MeOH | Gas-Phase | 2-Butanol | 1-Butanol |
|---|---|---|---|
| Hot utility | 0 | 6668 | 3912 |
| Cold utility | 2960 | 15,313 | 9604 |
| Heat integrated within process | 5104 | 5376 | 4047 |
| Waste heat generated | 2697 | 4048 | 2366 |
| Electricity | 624 | 683 | 625 |

The gas-phase process did not require hot utility as the reaction heat was sufficient to supply all process heating. Due to the combustion of purge streams for waste heat, the process produced a net heat output of 2.7 MW. In contrast, the liquid-phase process required external heat due to the more energy-intensive separation stage. In addition, less heat was available from the reactor due to the energy consumed in solvent evaporation, which is a downside to the efficient heat control provided by the liquid. Due to the less energy-intensive distillation stage, the use of the less volatile solvent 1-butanol was found to improve the energy efficiency of the liquid-phase process compared to 2-butanol significantly. The net heating duty in the 1-butanol process was 41% and the cooling duty 37% lower compared to the 2-butanol process. The utilization of waste heat significantly reduced the amount of external hot utility required: waste heat provided 38% of the process heat in each of the liquid-phase processes.

The electricity consumption of the liquid-phase processes was slightly higher compared to the gas-phase process. The increased per-pass conversion in the liquid-phase processes led to lower flow rate of the gas recycle, resulting in a reduction in electricity consumption by the recycle compressor. These reductions were, however, offset by the requirement of the solvent recycle pump in the liquid

phase processes. Energy requirements of the individual process equipment can be found in Table S9 in the Supplementary Material.

### 3.3. Environmental Analysis

The $CO_2$ and water balances of the alternative processes are presented in Table 7. A negative $CO_2$ balance signifies that the amount of $CO_2$ consumed in methanol synthesis was higher than the sum of the direct and indirect $CO_2$ emissions of the process. The lowest net emissions (−3046 kg/h) were found with the gas-phase process, in which all process heat was supplied by the exothermic reaction and fuel combustion was not required. In the liquid-phase process, generation of process heat lead to significant $CO_2$ emissions. In the 1-butanol process, the net emission was −1239 kg/h, or 59% higher compared to the gas-phase process. In the 2-butanol process, the net emission was positive (216 kg/h), as the amount of $CO_2$ emitted in heat generation was significantly higher compared to the 1-butanol process. In each process, waste heat generation led to additional emissions. However, in the liquid-phase processes, these emissions were offset by the reduced amount of natural gas burned for process heating.

**Table 7.** $CO_2$ and water balance of the alternative $CO_2$ hydrogenation to methanol processes.

|  | Gas-Phase | 2-Butanol | 1-Butanol |
| --- | --- | --- | --- |
| **$CO_2$ balance, kg/h** |  |  |  |
| Inlet streams | −3882 | −3882 | −3882 |
| Outlet streams | 560 | 369 | 255 |
| Hot utility (natural gas) | 0.0 | 2777 | 1837 |
| Waste heat combustion | 170 | 836 | 448 |
| Electricity (grid) | 106 | 116 | 106 |
| Net emissions | −3046 | 216 | −1239 |
| **Water balance** |  |  |  |
| Cooling water input, t/h | 379 | 516 | 516 |
| Solvent/water waste, kg/h | 1371 | 1574 | 1644 |
| Alcohol in waste, wt% * | 1% | 9% | 9% |

\* Methanol in the gas-phase process, solvent in the liquid-phase processes.

Although large amounts of cooling water were required in the processes, the environmental impact of the cooling water input was not particularly significant, assuming that the used water (at 25 °C) can be released without treatment and that a sufficient amount of water is readily available. Waste water treatment was simplified by minimizing the amount of mixed water/alcohol waste and by minimizing the alcohol content in the waste stream. The gas-phase process appeared the most favorable in terms of waste water treatment as the amount of waste produced was the lowest and the alcohol content (methanol) was only 1.1%. The waste water flow rate and alcohol content were similar in both the 2-butanol and 1-butanol processes.

As grid electricity was assumed to be used in the methanol synthesis processes, the environmental impact of electricity consumption was dependent on the sources of electricity supporting the grid at the particular location and time. The electricity consumption of the processes is compared in Table 6, and discussed in Section 3.2. The electricity consumption did not significantly differ between the alternative processes. At the $CO_2$ intensity of 170 g $CO_2$/kWh, the corresponding $CO_2$ emissions were 106 kg/h, 116 kg/h, and 106 kg/h for the gas-phase, 1-butanol, and 2-butanol processes, respectively. These emissions were not significant for the overall $CO_2$ balance, as seen in Table 7.

### 3.4. Methanol Production Cost and Net Present Value

The overall methanol production cost consisted of the variable and fixed operating costs and the annualized capital investment. A comparison of the production costs of the different processes is presented in Figure 5, while a detailed overview of the capital and operating costs is given in

the Supplementary Material (Tables S10 and S11). The gas-phase process was found to be the most competitive with a methanol production cost of 963 €/t. The overall production costs were higher with the liquid-phase processes due to the energy-intensive separation stage and the added cost of solvent make-up. The cost of 1205 €/t with the 1-butanol process was 25% higher compared to the gas-phase process, while the cost of 1349 €/t with the 2-butanol process was 40% higher compared to the gas-phase process.

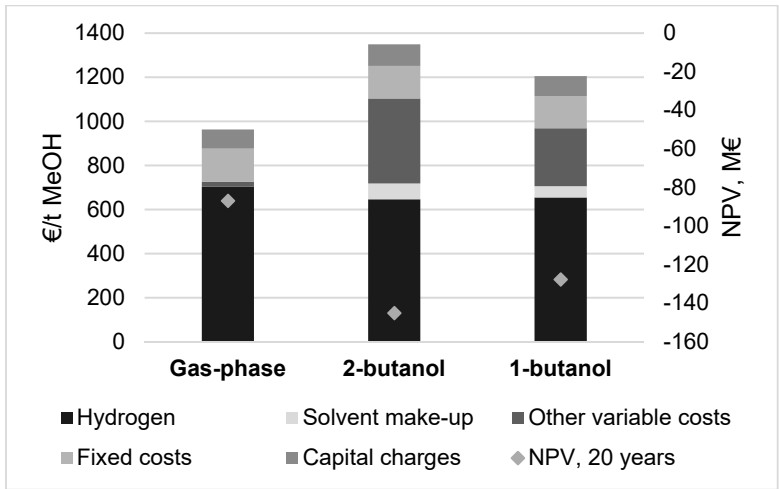

**Figure 5.** Methanol production cost and net present value (NPV) of the different $CO_2$ hydrogenation processes. Capital costs are annualized at an interest rate of 5% to obtain the annual capital charges. NPV is calculated assuming plant lifetime of 20 years and discount rate of 8%. Other variable costs: carbon dioxide, by-products and waste, and utilities.

The net present value (NPV) of each process over the assumed plant lifetime of 20 years is presented in Figure 5. The NPV was negative for all of the processes; thus, no process was financially feasible under the presented conditions and assumptions. In accordance with the lowest methanol production cost, the highest NPV (−87.0 M€) was found with the gas-phase process, whereas the NPV was −127.7 M€ for the 1-butanol process and −145 M€ for the 2-butanol process.

Figure 6 presents the contribution of individual variable cost components in the gas-phase and liquid phase processes. In all three processes, hydrogen was the main cost component, constituting 97%, 59%, and 68% of the variable costs, and 73%, 48%, and 54% of the total methanol production cost in the gas-phase, 2-butanol, and 1-butanol processes, respectively. In the gas-phase process, the cost of $CO_2$ was the second largest variable cost, while in the liquid-phase processes, the cost of utility steam was more significant due to the increased distillation energy requirement compared to the gas-phase process. The cost of heating in the 1-butanol process corresponded to 174 €/t MeOH, and if the heat requirement could be eliminated by process or site heat integration, the resulting methanol production cost would equal 1031 €/t. As the gas-phase process produced an output of heat that was available for steam generation, the variable costs were decreased by the steam credit. The steam credit also offset the electricity and cold utility costs of the process. In the 1-butanol process, the cost of solvent make-up constituted 5% of the variable costs and 4% of the overall cost. In the 2-butanol process, solvent make-up corresponded to 6% of the variable costs and 5% of the overall cost.

Compressors constituted the largest fraction of installed equipment costs, as seen in Figure 7. The total capital investment, calculated from the installed equipment costs by the factorial method, was lowest in the gas-phase process at 10.5 M€. The 2-butanol and 1-butanol processes required a capital investment of 12.9 M€ and 11.6 M€, respectively. The share of the separation section (flash vessels and distillation columns) was larger in the liquid-phase process compared to the gas-phase process. The higher capital costs of the 2-butanol process compared to the 1-butanol process were explained by the more costly distillation columns.

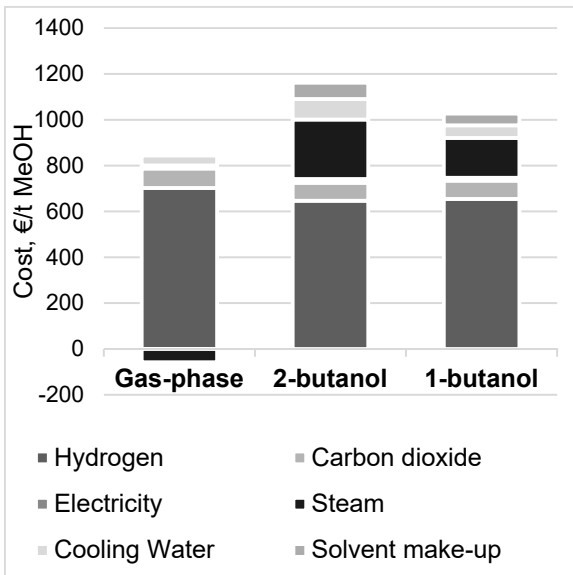

**Figure 6.** Distribution of the variable production costs in three $CO_2$ hydrogenation to methanol processes: the gas-phase process, the liquid-phase process with 2-butanol solvent, and the liquid-phase process with 1-butanol solvent.

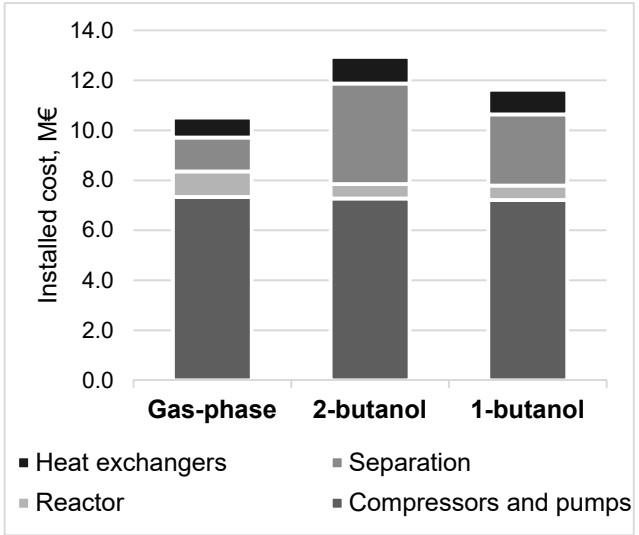

**Figure 7.** Installed equipment costs by type in three $CO_2$ hydrogenation to methanol processes: the gas-phase process, the liquid-phase process with 1-pentanol solvent, and the liquid-phase process with 1-pentanol solvent and solvent recovery by decantation. Separation included distillation, flash, and decanter vessels.

*3.5. Sensitivity Analysis*

A sensitivity analysis was performed in order to measure the effect of key variables on the economics of the various methanol synthesis processes. The sensitivity analysis was performed in terms of the methanol production cost, and NPV was not considered in the analysis. The cost of hydrogen, constituting a major fraction of the overall cost, was included in the sensitivity analysis. In addition, the effects of the costs of oxygen, $CO_2$, and the total capital investment were also analyzed. The gas-phase process and the 1-butanol process were selected for the sensitivity analysis. The cost of solvent was also included as a variable for the 1-butanol process. The results for the gas-phase process are shown in Figure 8, and those for the 1-butanol process are shown in Figure 9.

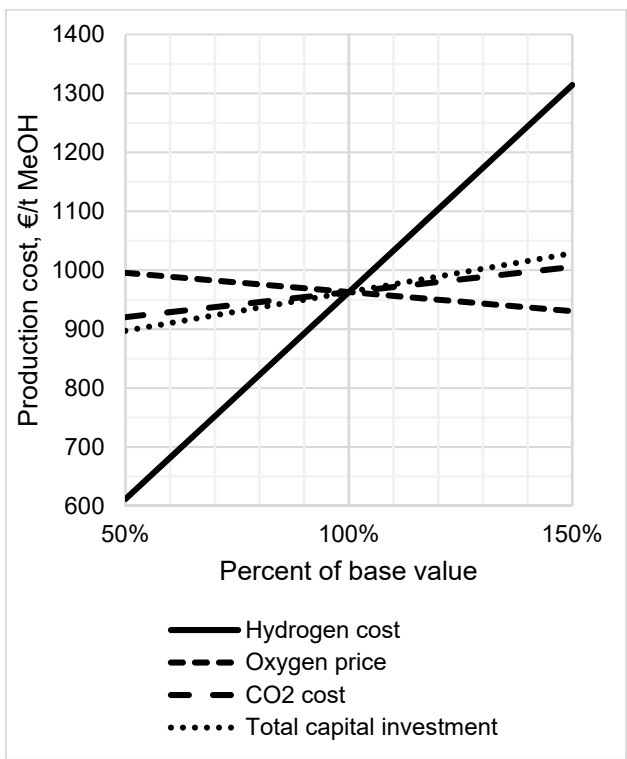

**Figure 8.** Sensitivity of methanol production cost to variations in selected parameters in the gas-phase methanol synthesis process. Base values: hydrogen cost 3000 €/t, oxygen price 70 €/t, $CO_2$ cost 50 €/t, total capital investment 17.9 M€.

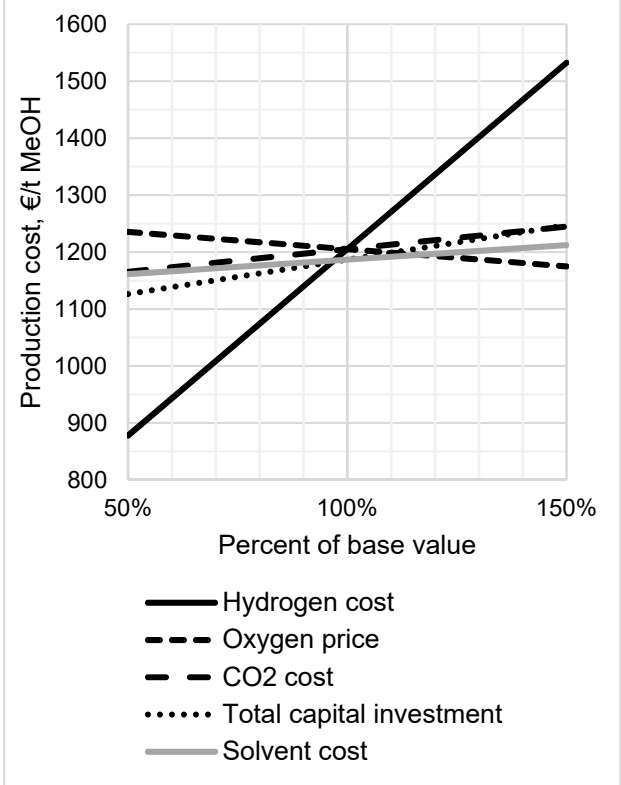

**Figure 9.** Sensitivity of methanol production cost to variations in selected parameters in the liquid-phase methanol synthesis process with 1-butanol solvent. Base values: hydrogen cost 3000 €/t, oxygen price 70 €/t, $CO_2$ cost 50 €/t, total capital investment 19.4 M€, solvent cost 500 €/t.

As expected from Figure 6, the cost of hydrogen was found to have the highest impact on the overall cost. In the gas-phase process, the overall production cost was 612 €/t at a hydrogen cost 50% of the base value, corresponding to a hydrogen cost of 1500 €/t. In the 1-butanol process, the production cost was 878 €/t at the hydrogen cost of 1500 €/t. The cost of hydrogen generated by water electrolysis was expected to decrease with the development of electrolyzer technology and the decreasing cost of renewable electricity. The decline in the cost of photovoltaic electricity has been particularly quick [50], potentially providing a more competitive route to renewable methanol compared to wind electricity in the near future. At the same time, the electrolyzer capital costs are expected to decrease, particularly in the case of more advanced proton exchange membrane (PEM) and solid oxide (SOEC) electrolyzers [51]. Clearly, a significant reduction in the hydrogen cost would be required to make these processes competitive at present methanol prices. The hydrogen cost required to reach a methanol cost of 400 €/t was approximately 600 €/t for the gas-phase process. For the 1-butanol process, this threshold was not reached even at zero hydrogen cost.

Due to the high contribution of operating costs to the overall production cost, the economics of $CO_2$ hydrogenation to methanol were not particularly sensitive to the capital investment. The economic impact of oxygen sales was apparent. If oxygen was not sold at all, the methanol production cost increased to 1028 €/t in the gas-phase process and 1266 €/t in the 1-pentanol process. The impact of oxygen sales was found to be higher than that of the $CO_2$ cost. Similar to the hydrogen cost, the cost of $CO_2$ capture is expected to decrease with technological development. However, the economics of the $CO_2$ hydrogenation to methanol process do not seem to be heavily affected by these developments.

*3.6. Summary*

The benefits and challenges of the liquid-phase $CO_2$ hydrogenation to methanol process compared to the gas-phase process are qualitatively summarized in Table 8. The advantage of the liquid-phase process was the increased per pass conversion due to the lower reaction temperature and the apparent favorable distribution of products between the gas and liquid phases in the reactor. At the same reaction pressure of 50 bar, the per pass $CO_2$ conversion was increased from 20% in the gas-phase process to 79% in the 2-butanol process and 81% in the 1-butanol phase process. The conversion in the liquid-phase processes was higher than the single-phase equilibrium conversion at the reaction conditions. This is presumably due to the beneficial phase equilibrium, i.e., the evaporation of products from the reacting liquid phase. However, uncertainties in the prediction of the phase distribution were present due to a lack of experimental data at the reaction conditions. In addition, whereas the gas-phase reaction could be modelled in detail using an available kinetic model, the liquid-phase reactions were modelled as equilibrium reactions without considering the reaction kinetics. In practice, the reaction rates in the liquid-phase process should be similar in magnitude to the gas-phase process in order to avoid excessively large reactor volumes.

In the gas-phase process, large amounts of gases are recycled in the reactor loop and purged from the process. As a result, the overall methanol yield was only 81%. In the liquid-phase processes, the overall methanol yield was 82% with 2-butanol and 88% with 1-butanol. However, the introduction of the solvent led to a more complicated overall process due to the complex phase equilibrium and mutual solubility of the reactants, products, and solvent, as well as increased demands on the separation stage. The large amounts of solvent present led to increased capital and energy intensity of the separation stage. The difference was manifested in the overall energy balance of the gas-phase and liquid-phase processes. While the gas-phase process produced a net heat output, the liquid phase processes required a significant net heat input, together with increased cooling duties. However, the energy consumption of the liquid-phase processes could be improved by more rigorous heat integration.

The different alcohols used as solvents had a significant effect on the energy consumption and overall methanol production cost. Compared to 2-butanol, whose boiling point is similar to that of water, the heat efficiency of the process was significantly improved by using 1-butanol, which possesses a higher boiling point. Ideally, even higher boiling point solvents would be used to simplify

the separation, potentially allowing selective evaporation of methanol and water from the reactor. However, the solvent should also be sufficiently active in the alcohol-promoted reaction to obtain reasonable reaction rates at low reaction temperatures. Development of increasingly active catalyst systems could allow even lower reaction temperatures, further increasing the equilibrium conversion. For example, Chen et al. [52] demonstrated a heterogeneous cascade catalytic system for liquid-phase $CO_2$ hydrogenation to methanol at 135 °C, with the reaction promoted by ethanol. The beneficial effect of the phase distribution in the liquid-phase reactor to the equilibrium conversion level, found in the present process modelling work, should also be experimentally verified and further investigated. Another approach to increase the equilibrium conversion and reaction rate could be provided by the adsorption of water from the reaction mixture [53] or by in situ condensation of water and methanol [54].

**Table 8.** Summary of the potential benefits and challenges of the liquid-phase $CO_2$ hydrogenation to methanol process with alcoholic solvents compared to the gas-phase process.

| Benefits | Challenges | Comments and Outlook |
|---|---|---|
| Lower reaction temperature leads to higher equilibrium conversion and lower reactant recycle and losses | - | Reaction temperature could be further lowered with catalyst development (e.g., Chen et al. [52]) |
| - | Complicated and energy-intensive separation leads to higher overall production cost and less favorable energy and $CO_2$ balances | The amount of solvent recycle should be minimized by utilizing high-boiling alcohols and improved reactor design (e.g., reactive distillation?); energy consumption could be minimized by improved heat integration |
| - | Formation of azeotropic alcohol-water mixtures further complicates solvent separation and recovery | Solvent recovery improved by phase separation of water and higher alcohols |
| Liquid-phase reaction potentially allows improved reactor temperature control and catalyst stability | - | Previously demonstrated in liquid-phase methanol synthesis using inert solvents [24] |

Additional costs to the liquid-phase process were caused by the loss of solvent. Two percent of 2-butanol and 3% of 1-butanol and 2-butanol entering the reactor were lost in downstream processing in the present processes. The loss was explained by the alcohol forming azeotropic mixtures with water. Without the introduction of an additional, complicated separation sequence to break the azeotropes, a fraction of the solvent was necessarily lost with the waste water removed from the process.

Environmental analysis was performed in terms of the $CO_2$ and water balances and the electricity consumption of the studied processes. The net $CO_2$ balance of the gas-phase and 1-butanol processes was found to be negative, i.e., the processes consumed more $CO_2$ than was released. However, the $CO_2$ balance was positive for the 2-butanol processes. The lowest net $CO_2$ emission, −3.0 t/h, was found with the gas-phase process, which did not require fuel combustion for heat generation. The hot utility requirement of the liquid-phase processes led to increased emissions. The net emission was −1.2 t/h with the 1-butanol process and 216 t/h with the 2-butanol process. The impact of the $CO_2$ emitted in electricity generation was insignificant as regards the process $CO_2$ balances. The gas-phase process produced the lowest flow rate of waste water (1371 kg/h), with a methanol content of 1.11 wt%. The 2-butanol process produced 1573 kg/h, and the 1-butanol process produced 1644 kg/h of waste water, both streams with an alcohol content of 9 wt%.

In terms of overall methanol production costs, the gas-phase process appeared more competitive than the liquid-phase processes, having a production cost of 963 €/t. This value appears to be consistent with previous studies. For example, Atsonios et al. [18], Rivera-Tinoco et al. [17], and Tremel et al. [23] estimated costs between 800 and 1000 €/t at production scales comparable to the present study. Pérez-Fortes et al. [16] estimated a break-even methanol price of 724 €/t for large industrial-scale production at a hydrogen cost of 3090 €/t (compared to 3000 €/t in the present study). However,

significantly lower costs have been reported by other authors [15,32,55,56]. For instance, Anicic et al. [15] found the cost of methanol produced from captured $CO_2$ and hydrogen from electrolysis (below 400 €/t) to be competitive with fossil fuel-based methanol production. It appears that this variation was a result of the different assumptions and methods employed in the analyses.

In the liquid-phase process, the production cost was 1205 €/t with 1-butanol and 1349 €/t with 2-butanol. The costs were 25% and 40% higher compared to the gas-phase process. The higher costs compared to the gas-phase process were explained by the more complex and energy-intensive separation stage and the added cost of the solvent make-up. The variation in the production cost for different liquid solvents was explained by the different capital costs and especially the energy costs of the distillation stage.

In all the processes considered here, the overall production cost was dominated by variable operating costs, especially the cost of hydrogen (73%, 48%, and 54% of the total production cost in the gas-phase, 2-butanol, and 1-butanol processes, respectively). As a result, the overall methanol production cost was highly sensitive to the cost of hydrogen. In the gas-phase process, a hydrogen cost of 600 €/t would be required to lower the methanol production cost to 400 €/t, representing the present methanol price. The liquid-phase processes did not meet this threshold value even at zero hydrogen cost. The heating costs in the 1-butanol process corresponded to 174 €/t MeOH. At present, none of the processes appeared economically feasible, with the 20-year net present values of −87 M€ for the gas-phase process, −128 M€ for the 1-butanol process, and −145 M€ for the 2-butanol process.

## 4. Conclusions

The feasibility of alternative $CO_2$ hydrogenation to methanol processes was compared by means of flowsheet modelling and economic analysis. The processes compared included a conventional gas-phase process and a liquid-phase process with 2-butanol and 1-butanol as alternative solvents. The potential benefit of the liquid-phase processes was that lower reaction temperatures were allowed by the co-catalytic activity of the alcoholic solvent. At the same reaction pressure of 50 bar, the per pass conversion of $CO_2$ in the reactor was increased from 20% in the gas-phase process to approximately 80% in the liquid-phase processes. The conversion in the liquid-phase processes was higher than the single-phase equilibrium conversion at the reaction conditions, apparently due to the evaporation of reaction products from the reacting liquid phase. As a result of the decreased amount of recycled gases, the overall conversion of $CO_2$ to methanol of 81% in the gas-phase process was increased to 82% and 88% in the 2-butanol and 1-butanol processes, respectively.

The benefits of the increased equilibrium conversion were found to be limited to slightly lower capital and operating costs in the reactor loop. However, the presence of large quantities of solvent in the reactor effluent led to a capital- and energy-intensive separation stage, which required two distillation columns compared to one in the gas-phase process. In addition, the formation of azeotropes between the alcohols and the water by-product complicated the separation and led to losses of solvent. Due to the increased energy consumption and the cost of solvent make-up, the methanol production cost of the most competitive liquid-phase process (1-butanol) was 1205 €/t, compared to the production cost of 963 €/t for the gas-phase process. The cost of heating in the 1-butanol process corresponded to 174 €/t MeOH, and if the heat requirement could be eliminated by process or site heat integration, the resulting methanol production cost would equal 1031 €/t.

None of the processes, in gas or liquid phases, were competitive at the present methanol price of approximately 400 €/t. The 20-year net present values of the gas-phase process and the 1-butanol and 2-butanol processes were estimated at −87 M€, −128 M€, and −145 M€, respectively. Hydrogen constituted the largest fraction of the overall cost in all processes, at 73%, 48%, and 54% in the above processes, respectively. In terms of $CO_2$ emissions, the most favorable $CO_2$ balance was found with the gas-phase process, with a net consumption of 3.0 t/h of $CO_2$. The 1-butanol process also showed a negative $CO_2$ balance, with a net $CO_2$ consumption of 1.2 t/h. Due to the higher amount of utility heat required, the 2-butanol process showed positive net emission of 0.2 t/h of $CO_2$. The water balance of

the gas-phase was also the most favorable, with the smallest amount of waste water released at the lowest alcohol content. The amount of and composition of waste water produced was similar in each of the liquid-phase processes.

To conclude, a feasible preliminary process design for the liquid-phase methanol synthesis process was developed. However, further optimization and improved heat integration are required to improve the process economics. Due to the greater complexity and less favorable energy balance of the liquid-phase process, the estimated methanol production cost was found to be 25% and 40% higher with the 1-butanol and 2-butanol processes compared to the baseline gas-phase process. It should be noted that significant assumptions and simplifications were made in modelling of the liquid-phase processes. In particular, reaction kinetics were not considered, and the equilibrium reactor model used presented a "best-case scenario" of the reactor performance. More detailed experimental information on reaction kinetics, yields, and thermodynamics is required to allow more in-depth modelling of the liquid-phase reaction system.

Regardless of these limitations, it is concluded that the liquid-phase $CO_2$ hydrogenation to methanol process shows potential, and the present findings provide valuable information for further development. The main advantage of the alcohol-promoted liquid-phase process is the higher equilibrium conversion allowed by operation at lower temperatures, and future development should aim to further increase the conversion levels to approach ideally full single-pass conversion. Possibilities include increasingly active catalyst/solvent systems allowing even lower reaction temperatures, removal of water from the reactor by means of adsorption, or in situ condensation of water and methanol from a gaseous reaction mixture. The apparent beneficial effect of the phase distribution in the liquid-phase reactor should also be investigated further, and both the co-catalytic effect and ease of separation should be considered in the selection of alcoholic solvents. More advanced reactor designs (e.g., reactive distillation) and separation methods (e.g., dividing wall columns) should also be considered, together with energy provision by heat pumps to decrease the cost and $CO_2$ intensity of the processes.

**Supplementary Materials:** The following are available online at http://www.mdpi.com/2227-9717/7/7/405/s1: Table S1: Stream table for the gas-phase $CO_2$ hydrogenation to methanol process. Table S2: Stream table for the liquid-phase $CO_2$ hydrogenation to methanol process with 2-butanol solvent. Table S3: Stream table for the liquid-phase $CO_2$ hydrogenation to methanol process with 1-butanol solvent. Table S4: Component mass balance of the gas-phase $CO_2$ hydrogenation to methanol process. Table S5: Component mass balance of the liquid-phase $CO_2$ hydrogenation to methanol process in 1-butanol solvent. Table S6: Component mass balance of the liquid-phase $CO_2$ hydrogenation to methanol process in 2-butanol solvent. Table S7: Thermal duties (kW) in the gas-phase $CO_2$ hydrogenation to methanol process. Table S8: Thermal duties (kW) in the liquid-phase $CO_2$ hydrogenation to methanol processes. Table S9: Electric duties (kW) in the alternative $CO_2$ hydrogenation to methanol processes. Table S10: Installed equipment costs of the alternative $CO_2$ hydrogenation to methanol processes (M€). Table S11: Operating and total methanol production costs of the alternative $CO_2$ hydrogenation to methanol processes (k€/t MeOH)

**Author Contributions:** Conceptualization, H.N. and A.L.; data curation, H.N.; methodology, H.N.; supervision, A.L. and T.K.; writing, original draft, H.N.; writing, review and editing, H.N., A.L., and T.K.

**Acknowledgments:** Funding provided by the LUT REFLEX platform and the Lappeenranta-Lahti University of Technology Doctoral School is gratefully acknowledged.

**Conflicts of Interest:** The authors declare no conflicts of interest.

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
