# Peer review of "CO2 Hydrogenation to Methanol by a Liquid-Phase Process with Alcoholic Solvents: A Techno-Economic Analysis"

_processes, doi:10.3390/pr7070405_

Round 1
Reviewer 1 Report
1. This is a detailed report on the investigation of liquid-phase process with alcoholic solvents and its comparison with gas-phase process, for CO2 hydrogenation to methanol. Presentation is well organized with many details. The manuscript is useful although it is long and the liquid-phase process is currently unattractive.
2. Line 135: the pressure drop in each exchanger was set at 2% based on reference 31. Is this also followed for pressure drop in reboilers and condensers? Heuristics in design books give certain values/ranges (e.g., 0.3 bar) and not %.
3. Lines 143 and 405: particle density is given as 1.75 kg/m3. Check this value and/or units.
4. Line 152: does thermodynamic equilibrium refer to free energy minimization? Clarify the basis of RSCSTR block.
5. Line 180: are installed equipment cost functions taken from Towler and Sinnott or those within Aspen Plus? Aspen Plus can provide capital cost estimates. Hence, why not use cost estimates given by Aspen Plus (instead of those from Towler and Sinnott)?
6. Line 220: low-pressure steam cost is about 10% lower than that of medium-pressure steam. Hence, what is the reason for the same cost for these two types of steam? What is its effect on processes evaluated?
7. Line 233: cooling water outlet temperature of 25 C is quite low. It can be higher (say, 40 to 45 C). So, what is the reason for the assumed value of 25 C?
8. Figures 2 and 3 can be more informative and more useful by labelling inlet, important intermediate and outlet streams.
9. Table VI: what is the meaning of the value for ‘Heat integrated’?
10. The pdf has the following errors at many places: Error! Reference source not found. Also, for some reason, this reviewer could not find/access the supplementary material of this manuscript.
11. Lines 668 to 670: what are the possible reasons for significantly lower costs in the cited literature? Is it because of hydrogen cost used in their calculations?
12. Energy for separation can be decreased by using dividing-wall columns and heat pumps (mechanical vapor recompression). Discuss their potential for the liquid-phase processes studied.
Author Response
Our response to Reviewer 1 is given in the attached Word file.

Reviewer 2 Report
CO2 hydrogenation to methanol by a liquid-phase process with alcoholic solvents: a techno-economic analysis by Harri Nieminen, Arto Laari and Tuomas Koiranen.
The authors investigated and compared the feasibilities of the conventional gas-phase process and an alternative liquid-phase process employing butanol as solvent and catalyst by means of flowsheet modelling and economic analysis.
As the authors admitted that this study is only a preliminary assessment of the new route for methanol production and needs some improvements before it can be considered for publication.
The abstract lacks a clear aim of the study. Please provide it.
The use 1-pentanol, and 1-hexanol beside butanol was stated within the Materials and Methods section. Abstract shows only the alcohols of butanol. This is confusing as it becomes the coverage of four alcohols by the study.
Introduction has omitted some key publications, such as:
Matzen, M., Demirel, Y. (2016) Methanol and dimethyl ether from renewable hydrogen and carbon dioxide: Alternative fuels production and life-cycle assessment. Journal of Cleaner Production 139, 1068-1077.
Matzen, M., Alhajj, M., Demirel, Y. (2015) Chemical storage of wind energy by renewable methanol production: Feasibility analysis using a multi-criteria decision matrix. Energy 93, 343-353.
L. Chen, Q. Jiang, Z. Song and D. Posarac, "Optimization of methanol yield from a Lurgi reactor," Chemical Engineering Technology, vol. 34, pp. 817-822, 2011.
N. Shamsul, S. Kamarudin, N. Rahman and N. Kofli, "An overview on the production of bio-methanol as potential renewable energy," Renewable and Sustainable Energy Reviews, vol. 33, pp. 578-588, 2014.
K. A. Ali, A. Z. Abdullah and A. R. Mohamed, "Recent development in catalytic technologies for methanol synthesis from renewable sources: A critical review," Renewable and Sustainable Energy Reviews, vol. 44, pp. 508-518, 2015.
3 Introduction should be shortened to omit the repeated well-known text.
150-151: “Due to the lack of any detailed kinetic model for the alcohol promoted reaction route, the reactor was modelled with the RSCSTR block based on the thermodynamic equilibrium”
Which alcohol?
RSCSTR block requires kinetic data at its’ best. Is it RSTOIC?
162: The results of the reactor sizing are presented in Section Error! Reference source not found?
163: It is not clear if the ‘Carbon Tracking” feature of Aspen Plus with selected carbon emission standard (EU or EPA) and ultimate fuel. Please elaborate.
178: It is not clear if mapping, sizing and cost estimations by the Aspen Plus are used? Please elaborate.
216: It is not clear if a ‘utility block’ has been created within the Aspen Plus in order to estimate the rate and cost of utility production. Please elaborate.
It is also not clear how a previous cost is update to a year representing the present time based for cost calculations. Is 2018 or 2019 the year based for cost calculations?
260: “A discount rate of 8 % was assumed, and taxes and depreciation were not considered” This statement should be justified by addressing the neglecting tax and depreciation and a discount rate just 3% higher than that of interest rate.
285: The flowsheet of the gas-phase methanol synthesis process is presented in Error! Reference source not found..
319: presented in Error! Reference source not found.
384: “Error! Reference source not found. presents the temperature and composition profile of the”
Figure 4 shows that the most of the length of the reactor is redundant as the reactants and products mole fractions unchanged. Please elaborate it.
525: presented in Error! Reference source not found.. The NPV is negative for all of the processes; thus, no
572-573: Error! Reference source not found., and those for the 1-butanol process are shown in Error! Reference source not found..
582:As expected from Error! Reference source not found., the cost of hydrogen was found to have
Figures 6 and 7 do not include the use 1-pentanol, and 1-hexanol beside butanol. Please elaborate.
Feasibility primarily relies on NPV based economic analysis; however the recent trend combines sustainability metrics and economic indicators toward a comprehensive feasibility analysis. Please refer to”
K. Kowalski, S. Stagl, R. Madlener and I. Omann, "Sustainable energy futures: Methodological challenges in combining scenarios and participatory multi-criteria analysis," European Journal of Operational Research, vol. 197, no. 3, pp. 1069-1074, 2009.
S. Pugh, "Concept Selection: a Method that Works," in Proceedings International Conference on Engineering Design, Zurich, 1981.
Matzen, M., Alhajj, M., Demirel, Y. (2015) Chemical storage of wind energy by renewable methanol production: Feasibility analysis using a multi-criteria decision matrix. Energy 93, 343-353.
Author Response
The response to Reviewer 2 is given in the attached Word file.

Round 2
Reviewer 1 Report
The authors provided their responses to all comments of the reviewer and also revised the manuscript. The following comments are to increase the usefulness of the revised manuscript.
Is the pressure drop of 2% in each exchanger comparable to heuristics on this (e.g., 0.3 bar)? This should be clarified in the manuscript.
The reason, namely, transparency for the following addition should be given in the manuscript. “The capital costs were estimated by the factorial method according to Towler and Sinnott [37]. The installed equipment costs for the estimation were obtained from the cost functions integrated into the Aspen Plus software.”
The following two responses of authors should be included briefly in the revised manuscript.
“The cost of low-pressure steam was set equal to that of medium-pressure stream only for simplication. The impact of this is highest on the 2-butanol process, which consumed the highest amount of low-pressure steam. Setting the cost of low-pressure steam at 90 % of the cost of medium-pressure steam results in a methanol production cost decrease of approximately 10 €/t, or a 0.7 % decrease in the overall cost. As such we find this effect insignificant.”
“For the cooling water inlet and outlet temperatures, the default values for the Aspen Plus utility template for cooling water were used. Setting the outlet temperature to 40 C would result in a significant decrease in the cooling water usage (75 % decrease in the 2-butanol process). This would result in a decreased overall methanol cost by approximately 67 €/t, or 5 %, which is quite significant. However, this would not change the overall conclusion of the work, as the 1-butanol and gas-phase processes also consume significant amounts of cooling water, and the relative competitiveness would not be changed.”
Author Response
Is the pressure drop of 2% in each exchanger comparable to heuristics on this (e.g., 0.3 bar)? This should be clarified in the manuscript.
The pressure drop of 2% was selected based on the heuristic from the reference [30]. With a maximum process pressure of slightly above 50 bar, the highest pressure drop found is approximately 1 bar.
The reason, namely, transparency for the following addition should be given in the manuscript. “The capital costs were estimated by the factorial method according to Towler and Sinnott [37]. The installed equipment costs for the estimation were obtained from the cost functions integrated into the Aspen Plus software.”
This is now clarified in the manuscript, on lines 166-167.
“The cost of low-pressure steam was set equal to that of medium-pressure stream only for simplication. The impact of this is highest on the 2-butanol process, which consumed the highest amount of low-pressure steam. Setting the cost of low-pressure steam at 90 % of the cost of medium-pressure steam results in a methanol production cost decrease of approximately 10 €/t, or a 0.7 % decrease in the overall cost. As such we find this effect insignificant.”
This text was adapted and added to the manuscript on lines 210-214.
“For the cooling water inlet and outlet temperatures, the default values for the Aspen Plus utility template for cooling water were used. Setting the outlet temperature to 40 C would result in a significant decrease in the cooling water usage (75 % decrease in the 2-butanol process). This would result in a decreased overall methanol cost by approximately 67 €/t, or 5 %, which is quite significant. However, this would not change the overall conclusion of the work, as the 1-butanol and gas-phase processes also consume significant amounts of cooling water, and the relative competitiveness would not be changed.”
This text was adapted and added to the manuscript on lines 227-232.
Reviewer 2 Report
I think the authors have revised the original manuscript with useful analysis to the readers.
Author Response
The comments received from the Reviewer are gratefully acknowledged.